# Deciphering Molecular Factors That Affect Electron Transfer at the Cell Surface of Electroactive Bacteria: The Case of OmcA from *Shewanella oneidensis* MR-1

**DOI:** 10.3390/microorganisms11010079

**Published:** 2022-12-28

**Authors:** Ricardo O. Louro, Giovanni Rusconi, Bruno M. Fonseca, Catarina M. Paquete

**Affiliations:** Instituto de Tecnologia Química e Biológica António Xavier, Universidade Nova de Lisboa, 2780-157 Oeiras, Portugal

**Keywords:** *Shewanella oneidensis* MR-1, outer-membrane cytochrome, extracellular respiration, indirect electron transfer, mutagenesis, binding processes

## Abstract

Multiheme cytochromes play a central role in extracellular electron transfer, a process that allows microorganisms to sustain their metabolism with external electron acceptors or donors. In *Shewanella oneidensis* MR-1, the decaheme cytochromes OmcA and MtrC show functional specificity for interaction with soluble and insoluble redox partners. In this work, the capacity of extracellular electron transfer by mutant variants of *S. oneidensis* MR-1 OmcA was investigated. The results show that amino acid mutations can affect protein stability and alter the redox properties of the protein, without affecting the ability to perform extracellular electron transfer to methyl orange dye or a poised electrode. The results also show that there is a good correlation between the reduction of the dye and the current generated at the electrode for most but not all mutants. This observation opens the door for investigations of the molecular mechanisms of interaction with different electron acceptors to tailor these surface exposed cytochromes towards specific bio-based applications.

## 1. Introduction

Electroactive organisms possess extracellular electron transfer (EET) capabilities that enable them to be used in microbial electrochemical technologies (METs), including microbial fuel cells (MFC) and microbial electrosynthesis (MES), to produce bioenergy and added-value compounds, respectively. Given the interest in these technologies as promising sustainable processes for wastewater treatment, biosensing, bioremediation, and the production of biofuels [1,2,3,4], there has been a growing interest in understanding the EET processes performed by electroactive bacteria. This information was already shown to be crucial for the optimization of bioreactors and electrode materials, as well as to engineer or tailor electroactive organisms [5,6,7]. The use of genetic engineering strategies to develop electroactive bacteria with enhanced EET properties and increase current generation in MFC has increased in the last decade. Genetic engineering has been mainly used to improve indirect electron transfer [8], increase substrate oxidation [9,10,11], enhance biofilm formation [12,13,14,15], as well as increase the expression of multiheme cytochromes, the key components of EET processes [16,17,18].

*Shewanella oneidensis* MR-1 (SOMR1), a mesophilic Gram-negative facultative anaerobic bacterium renowned for respiratory versatility, is one of the most extensively studied electroactive organisms [19]. In this bacterium, EET processes are performed by a repertoire of multiheme cytochromes, including the inner-membrane tetraheme cytochrome CymA, the periplasmic cytochromes STC and FccA, and the decaheme cytochromes MtrA and MtrC that are part of the outer-membrane porin complex MtrCAB [20]. These cytochromes form a conductive pathway that enables electrons derived from the oxidation of electron donors (e.g., lactate) to be transferred outside of the cell for the reduction of electron acceptors (e.g., insoluble metal oxides in their natural environment and electrodes in METs). In this pathway, CymA receives electrons from the quinone pool and transfers them either to STC or FccA. These proteins can then transfer the electrons to the decaheme cytochrome MtrA that is inserted in the MtrB porin to deliver them to the terminal reductase MtrC [20,21]. Interaction studies have demonstrated that FccA does not interact with STC, mainly as a consequence of their negative surface charges, suggesting that two non-mixing redox pathways co-exist in the periplasmic space of SOMR1 [22]. SOMR1 also contains the decaheme cytochrome OmcA, a lipoprotein proposed to function as a terminal reductase, receiving electrons from MtrCAB complex and transferring them to electron shuttles or solid electron acceptors. Outer-membrane cytochromes, including MtrC and OmcA, were shown to engage in lateral diffusion at the outer-membrane to define the path for electron transport, enabling a long-distance conduction mechanism [23].

The crystal structure of OmcA [24] was crucial to provide insights into the distinct functions of the different hemes. While heme 5 was proposed to interact with MtrC, hemes 2 and 7, located near flavin binding domains [25,26], were proposed to interact with electron shuttles [27]. Docking studies revealed that anthraquinone 2,6-disulfonate (AQDS) is indeed found in the close vicinity of hemes 2 and 7, with heme 2 being the one that presented the lowest energy conformation ensemble for the complex [27]. Hemes 9 and 10, located near a proposed hematite binding motif with the conserved sequence Ser/Thr-Pro-Ser/Thr [28], were proposed to interact with minerals and metal ions. 

Site-directed mutagenesis was recently used to study the functional specificity of several of the hemes in OmcA [29], with the aim of using genetic engineering to enhance electron transfer processes. It was demonstrated that while the modification of the distal axial ligand of heme 2 from a histidine to a methionine did not significantly affect the reactivity of OmcA with FMN, the same type of replacement on heme 7 significantly changed the overall oxidation of OmcA by redox shuttles and weakened the binding affinity for FMN [29]. This difference was explained by the fact that heme 7 is near a disulfide bond whose redox state has been proposed to control the changes of OmcA towards a flavocytochrome state [30]. Amino acid mutations can lead to modifications in the three-dimensional structure of the protein, facilitating unfolding or loss of function, or, in the case of redox proteins, change their redox properties [31]. In this work, we used site-directed mutagenesis to modify the distal axial ligands of all the hemes in OmcA and evaluate, by in vitro studies, their redox properties, and the impact in reducing soluble electron shuttles. Furthermore, the ability of *S. oneidensis* to use these different OmcA variants in performing EET to different electron acceptors was explored. This work complements the previous work performed by Neto et al. [29], with the main aim of understanding the factors that control electron transfer at the cell-surface of electroactive organisms to be able to tailor electroactive organisms towards improved properties. The data obtained showed that amino acid modifications in OmcA can affect the reactivity of the protein and affect electron transfer to extracellular electron acceptors in living cells. This work demonstrates that although several factors influence electron transfer to electrodes, the binding process is one of the most important in defining EET. These results open the door for the rational manipulation of terminal electron acceptors in electroactive organisms, expecting to improve electron transfer at the electrode–microbe interface. 

## 2. Materials and Methods

### 2.1. Construction of Plasmids Harboring OmcA Mutants

The plasmid used to produce OmcA mutant protein for the in vitro studies was pLS147 provided by Dr. Liang Shi. This plasmid, based on the commercially available plasmid pBAD202/D-TOPO, contains a modified version of the *omcA* gene, where the signal peptide was replaced by the signal peptide of MtrB from SOMR1 to produce a soluble version of the protein [24]. A histidine tag was available at the C-terminal to facilitate the purification process [24]. The distal histidine ligand of hemes 1, 3, 4, 6, and 8 was mutated to a methionine using site-directed mutagenesis as previously described [29]. The pBAD202/D-TOPO plasmids containing the different OmcA variants were transformed in wild-type SOMR1.

For the in vivo studies, native OmcA was first cloned in pBBR1MCS-2 vector using NEBuilder^®^ HiFi DNA Assembly and the primers pBBR_OmcA_Forw and pBBR_OmcA_Rev (Table 1). The plasmid pBBR1MCS-2 was a gift from Kenneth Peterson [32]. For the production of OmcA mutants in this plasmid, two different approaches were used: site-directed mutagenesis using the pBBR1MCS-2 vector containing native OmcA (for mutants H1, H3, H6, H7, and H8); and NEBuilder^®^ HiFi DNA Assembly using mutated OmcA genes previously cloned in pBAD202/D-TOPO vector (for mutants H2, H4, H5, H9, and H10). In this case, the mutated genes were amplified using the primers pBBR_OmcA_Forw and pBBR_OmcA_Rev from the plasmid harboring each mutated gene. For the mutant H9, the primer pBBR_OmcA_Rev was replaced by the primer pBBR_OmcA_H9_Rev to guarantee the insertion of the mutation. The pBBR1MCS-2 plasmids containing the different versions of OmcA were transformed in SOMR1 ΔOmcA ΔMtrC. The use of this double knock-out strain is because of the overlapping role of both OmcA and MtrC [33]. To guarantee that the data obtained is due to the mutation and not to any other interacting factor, SOMR1 ΔOmcA ΔMtrC was used for all the in vivo experiments, including the negative (with empty pBBR1MCS-2 plasmid) and positive (with pBBR1MCS-2 plasmid containing native OmcA) control. 

All the primers used in this study are listed in Table 1 and in [29], while all the plasmids used are presented in Table 2. All the constructs were confirmed by DNA sequencing (Eurofins, Germany). The transformation was achieved using electroporation [34,35]. The pBAD202/D-TOPO plasmid harboring native *omcA* gene and the SOMR1 ΔOmcA ΔMtrC strain were kindly provided by Professor Johannes Gescher from Hamburg University of Technology, Germany.

### 2.2. Purification of OmcA Mutants

The mutant proteins OmcA_H1, OmcA_H3, OmcA_H4, OmcA_H6, OmcA_H8, OmcA_H9, and OmcA_H10 where the distal histidine ligand of the respective heme has been modified to a methionine, were produced and purified as previously described [29]. The purity of the proteins was verified by a single band in the SDS-PAGE and by an A_408_/A_280_ ratio of above 5 measured by UV-visible spectroscopy. All the proteins were washed with 20 mM phosphate buffer, 100 mM KCl at pH 7.6. This buffer was used for all experiments. The concentration of the proteins was determined by UV-visible spectroscopy using a ε_408 nm_ of 125,000 M^−1^ cm^−1^ per heme for the oxidized state of the cytochrome. ^1^H-1D-NMR spectra were collected for the different mutants on a Bruker Avance II+ 500 MHz NMR spectrometer equipped with a 5 mm TCI C/N prodigy cryoprobe. These experiments were performed at 25 °C.

### 2.3. Cyclic Voltammetry of Native OmcA and OmcA Mutants 

Cyclic voltammetry (CV) was performed using a three-electrode system cell configuration consisting of a pyrolytic graphite edge (PGE) working electrode (IJ Cambria Scientific, Llanelli, UK), an Ag/AgCl (3M KCl) reference electrode, and a graphite rod counter electrode (IJ Cambria Scientific, Llanelli, UK). The experiments were performed inside an anaerobic chamber (Coy Laboratory Products) at 25 °C controlled by an external bath. Before use, the PGE electrode was polished with aqueous Al_2_O_3_ slurry (1.0 μm), rinsed with water, and dried with a tissue before being exposed to the protein. For the experiments, 2 μL of the protein (concentration between 24 and 220 μM) was deposited onto the PGE electrode and left to dry. CV experiments were performed at a scan rate of 100 mV/s using CHI software. All potentials are reported with respect to Standard Hydrogen Electrode (SHE) by the addition of 210 mV [36] to those measured. QSOAS (version 1.0) [37] was used to subtract the capacitive current of the raw electrochemical data.

### 2.4. Kinetic Experiments with Electron Shuttles

To explore the ability of the OmcA mutants to perform indirect electron transfer to different electron shuttles, kinetic experiments using a stopped-flow apparatus (HI-TECH Scientific SF-61 DX2) installed inside an anaerobic chamber (M-Braun 150) containing less than 5 ppm of oxygen were performed [27,29]. Four electron shuttles were tested: AQDS, flavin mononucleotide (FMN), riboflavin (RF), and phenazine methosulfate (PMS). These experiments were performed at 25 °C, and all the solutions were prepared inside the anaerobic chamber using degassed buffer (20 mM phosphate buffer, 100 mM KCl at pH 7.6). The concentrations of the proteins were determined by UV-visible spectroscopy using a ε_552 nm_ of 30,000 M^−1^ cm^−1^ per heme, for the reduced state of the cytochrome [38]. The concentration of the electron shuttles was determined by UV-visible spectroscopy using ε_445 nm_ of 12,500 M^−1^ cm^−1^ for RF [39], ε_445 nm_ of 12,200 M^−1^ cm^−1^ for FMN [40], ε_326 nm_ of 5200 M^−1^ cm^−1^ for AQDS [41], and ε_387nm_ of 26,300 M^−1^ cm^−1^ for PMS [42].

To perform the kinetic experiments, reduced mutants OmcA_H4, OmcA_H6, and OmcA_H8, prepared with the addition of small volumes of a concentrated solution of sodium dithionite, were mixed with the different electron shuttles. Data were collected by measuring the light absorption changes at 552 nm as previously described [27]. For each mutant, the fully oxidized and fully reduced state of the protein were obtained by calibration with potassium ferricyanide and sodium dithionite, respectively. Data analysis was performed as previously explained [27].

### 2.5. Interactions Studies with FMN Using NMR

Interaction studies between FMN and OmcA mutants were performed as previously described for OmcA and mutants OmcA_H4, OmcA_H5, OmcA_H6, OmcA_H8, OmcA_H9, and OmcA_H10 [27,29]. Briefly, 100 μM FMN samples were titrated against increasing amounts of the target mutant protein and ^31^P-1D-NMR spectra were recorded after each addition [27]. The NMR experiments, performed at 25 °C, were acquired on a Bruker Avance II 500 MHz NMR spectrometer equipped with a SEX probe. ^31^P-1D-NMR experiments were collected with proton decoupling and calibrated using phosphate buffer as an internal reference. Data analysis and binding affinities determination were performed as previously described [29].

### 2.6. Reduction of Methyl Orange by S. oneidensis 

The ability of native and mutant OmcA to perform EET was evaluated through the decolorization of methyl orange using living cells as previously described [43]. The experiments were performed in triplicate for each strain (e.g., SOMR1 ΔOmcA ΔMtrC strains carrying the pBBR1MCS-2 plasmid with mutated OmcA), using a 96-well plate with a flat bottom. Briefly, bacterial cells grown overnight in LB medium at 30 °C and 150 rpm were inoculated in SBM minimal medium supplemented with lactate (20 mM) and methyl orange (50 μM), previously de-aerated with N_2_ for 15 min [43].

Decolorization of methyl orange was followed over time at 465 nm at 30 °C using a microplate spectrophotometer (Multiskan Sky Microplate Spectrophotometer from Thermo Scientific, Waltham, MA, USA). The preparation of the plate was conducted inside an anaerobic chamber (Coy Laboratory Products), and to maintain anaerobic conditions during the experiment, de-aerated Johnson oil was added to each well prior the sealing of the plate with a disposable seal and lid. SOMR1 ΔOmcA ΔMtrC carrying pBBR1MCS-2 (SOMR1 ΔOmcA ΔMtrC/pBBR_empty) and the wild-type *omcA* gene cloned in this plasmid (SOMR1 ΔOmcA ΔMtrC/pBBR_OmcA) were used as controls. These experiments were repeated at least two times independently, and the results were reproducible between the strains.

### 2.7. Reduction of Electrodes by S. oneidensis

The electroactivity of SOMR1 ΔOmcA ΔMtrC containing pBBR1MCS-2 and carrying the gene for the different mutant variants of OmcA (SOMR1 ΔOmcA ΔMtrC/pBBR_OmcA H1-H10) was evaluated by the ability of these strains to reduce screen-printed electrodes (SPEs), using a similar strategy as described previously for *Geobacter sulfurreducens* [44]. Toward this, cells from the different strains, grown overnight in LB medium at 30 °C and 150 rpm, were harvested at 12,000 rpm for 1 min and resuspended in SBM minimal medium supplemented with lactate (20 mM) to achieve an OD_600nm_ between 0.8–1.0. Then, after de-aeration with N_2_ for 15 min, 1 mL of *S. oneidensis* cell suspension was added to the SPEs, using a sealed cap-tube inverted fixed in the SPEs electrode with hot glue (see Appendix A). All electrochemical assays were performed in a SPEs C11L (Dropsens, Spain) that is composed of a three-electrode configuration with a working electrode of carbon ink (surface 0.126 cm^2^), a carbon counter electrode, and an Ag/AgCl reference electrode. The current of the chronoamperometry assays was measured every 120 s, and a fixed potential of 0.2 V was used for all the experiments. The experiments for each strain were performed at least in duplicate, and in each set of experiments, SOMR1 ΔOmcA ΔMtrC carrying pBBR1MCS-2 (SOMR1 ΔOmcA ΔMtrC/pBBR_empty) and this plasmid carrying the wild-type *omcA* gene (SOMR1 ΔOmcA ΔMtrC/pBBR_OmcA) were used as controls.

## 3. Results

### 3.1. Not All OmcA Protein Mutant Variants Retain the Native Overall Structure 

The replacement of the distal ligand of the hemes from histidine to methionine generally changes the reduction potential of the heme to more positive values [45] and changes the electronic structure of the heme orbitals [46], but retains the coordination number and the spin-state of the heme unless steric hindrance prevents methionine to bind to the iron. In a previous work, the distal ligand of hemes 2, 5, 7, 9, and 10 of OmcA has been modified to a methionine [29], and in this work, the modification was achieved for the remaining hemes. OmcA with the methionine as the distal ligand of hemes 4, 6, and 8 could be produced in high amounts (>10 mg/L culture), while OmcA with hemes 1 and 3 mutated could only be obtained in low amounts (<0.1 mg/L culture). An SDS-PAGE gel stained for *c*-type heme proteins [47] showed that *S. oneidensis* is able to produce mutants OmcA_H1 and OmcA_H3 (Appendix A), but the amount is significantly lower than that of the other mutants or native OmcA. The 1D ^1^H NMR spectrum of the downfield paramagnetically shifted signals of oxidized, low-spin heme proteins is dominated by heme methyl signals. Figure 1 shows that the spectra of mutants harboring a methionine as the distal ligand of heme 1 and heme 3 are severely disturbed, suggesting either partial unfolding of the protein or the co-existence of multiple conformations, with the lower stability and yield of these variants. This could be explained by the position of the distal ligand of both hemes that are close to each other (Appendix A), suggesting that the coordination of these hemes is important to stabilize the folding of the protein. Given the low yield obtained for these OmcA mutants, these proteins were not studied further. 

Although the 1D ^1^H NMR spectrum of OmcA_H4 is similar to that of OmcA_H3, it is clear that the replacement of the histidine by methionine in heme 4 did not affect the stability of the protein, given the high amount of protein obtained for this mutant. This indicates that the structural modifications that occur in OmcA_H4 are different from those obtained for mutants OmcA_H1 and OmcA_H3 and do not affect the stability of the protein. 

For the mutants OmcA_H6 and OmcA_H8, most signals appeared in the same frequency as the native protein, with only a few signals being affected (Figure 1). Given the exquisite sensitivity of paramagnetic shifts to structural changes, these observations indicate that the overall folding of the protein was retained for these OmcA mutants and that the signals with major changes are likely from the heme for which the axial ligand was mutated (Figure 1). 

### 3.2. Mutations in the Axial Ligands of the Hemes Change the Reduction Potential of the Individual Redox Centers of OmcA

To evaluate the direct electron transfer of native OmcA and its mutants, protein film voltammetry was used. This technique allows the investigation of the reduction and oxidation processes of molecular species, providing information regarding their electron transfer mechanisms. The cyclic voltammograms of native OmcA and all the produced mutants are presented in Figure 2. Native OmcA titrates between 0 to −400 mV vs. SHE, which is in line with what was reported in the literature using UV-visible spectroelectrochemistry [48]. Although the mutations did not affect the overall potential window where the protein is electrochemically active (between 0 and −400 mV vs. SHE), the shape of the voltammogram is different between the mutants (Figure 2).

It is clear that the substitutions of the histidine axial ligand to a methionine affects the thermodynamic properties of OmcA. Up to date, the determination of the reduction potential of individual hemes has been only accomplished in multiheme cytochromes with up to six hemes [49]. This approach can only be performed for proteins where a discrimination of redox transitions of the individual hemes can be achieved, which is mainly achieved by NMR [50]. For OmcA, this discrimination has not yet been performed given the high number of hemes and size of the protein. For this reason, it is not possible to determine the heme(s) that was affected by the mutation, and if this was the one where the mutation was introduced, or if it was a nearby heme.

### 3.3. Mutation of the Axial Ligand of the Respective Heme Affects Electron Transfer Rates from OmcA to Soluble Acceptors

Kinetics of oxidation of OmcA mutants OmcA_H4, OmcA_H6, and OmcA_H8 by the electron shuttles FMN, RF, PMS, and AQDS were studied using stopped flow as previously described [27]. These four compounds represent the chemical and electrostatic diversity of electron shuttles that can be found by *S. oneidensis* in the environment [27,29]. The oxidation of the OmcA mutants with the different electron shuttles occurred in the millisecond timescale, with most of the reactions occurring within the dead time of the stopped flow (Figure 3). The extent of oxidation of the protein by each redox shuttle is determined by the value of the reduction potential of the shuttle versus the potentials of the various hemes. PMS has a positive reduction potential, which allows OmcA (native and mutants) to achieve nearly complete oxidation (i.e., reduced fraction of 0) because all hemes in OmcA have a lower potential and, therefore, PMS is capable of extracting electrons from all of them. Nonetheless, the replacement of a histidine for a methionine in heme 8 has led to an increase of 10% of residual reduction of OmcA which may indicate that this heme has been brought to a potential closer to PMS by the mutation. FMN and RF have similar reduction potentials and, therefore, oxidize OmcA and its mutants to the same extent, indicating that none of the mutations has given rise to a transition of the potential of one heme to a value that is above that of these two mediators. AQDS has a slightly higher reduction potential than FMN and RF. Interestingly, AQDS is able to oxidize OmcA mutated in heme 4 to less than 30%, which is lower than the 40% observed for wild-type protein [29], indicating that the mutation has lowered the potential of at least one heme from a value that is above that of AQDS in the native protein to one that is below in the mutant. The fact that the extent of reduction of OmcA achieved by FMN and RF remains the same as the native protein means that the affected heme had its potential lowered to a value between that of AQDS and FMN/RF. Replacement of a histidine for a methionine in the axial coordination of heme iron in the absence of other changes leads to an increase of the reduction potential due to the extra stabilization of the Fe(II) state by the coordinating sulfur versus the nitrogen. The fact that the opposite is observed strongly suggests that this mutation, although still compatible with a protein that has not lost its overall fold, has made one heme, which is not necessarily heme 4, more solvent exposed and, therefore, with lower potential. This functional observation is in line with the numerous changes observed in the NMR spectrum of mutant OmcA_H4 (Figure 1).

### 3.4. Mutations in OmcA Did Not Significantly Affect the Binding of FMN

The effect of the mutations on the hemes 4, 5, 6, 8, 9, and 10 in the binding of FMN was explored by ^31^P 1D-NMR as previously described [29]. For all the mutants tested, upon protein binding, the phosphorous atom signal shifts position and broadens, indicating an interaction between FMN and the protein in a fast regime on the NMR timescale (Appendix A). The fitting of the data with the binding model previously described [29] (Appendix A) shows that weak transient interactions occur between OmcA and FMN. The values of the dissociation constants of the different mutants (Table 3) are all typical of electron transfer reactions between cytochromes and their physiological partners [27,51]. Interestingly, the dissociation constant obtained for the mutants OmcA_H4, OmcA_H5, OmcA_H6, OmcA_H7, and OmcA_H8 indicate slightly weaker binding than the native OmcA, while for mutants OmcA_H2, OmcA_H9, and OmcA_H10, the value of the dissociation constants is more similar to that obtained for native OmcA (Table 2). Nonetheless, it is not expected that the differences observed would have a significant impact on the electron transfer processes, given that the values are all in the sub millimolar range.

### 3.5. The Electroactivity of the Different OmcA Mutants in S. oneidensis Generally Matches the Reactivity with Methyl Orange

The capacity of the *S. oneidensis* strains carrying different OmcA mutants in performing extracellular electron transfer was evaluated by the rate at which they decolorize methyl orange (Figure 4) and by the current produced in an electrode (Figure 5 and Appendix A).

It has been demonstrated that outer-membrane cytochromes play a key role in the reduction of methyl orange [52,53], given that this azo dye does not cross the outer-membrane of *Shewanella*. Indeed, the decolorization of methyl orange by the *S. oneidensis* strain lacking both OmcA and MtrC (pBBR_empty in Figure 4) occurs at a slower rate when compared with the strain containing native OmcA (pBBR_OmcA in Figure 4). This clearly shows that OmcA plays a significant role in the reduction of methyl orange at the cell surface of *S. oneidensis*. Given that the decolorization is not completely abolished when both MtrC and OmcA are absent from the cell surface, there must be other processes or other proteins that also contribute to the decolorization of this azo dye.

Most of the OmcA mutants decolorize methyl orange at the same rate as native OmcA. *S. oneidensis* containing OmcA_H1, OmcA_H3, and OmcA_H6 decolorized methyl orange at a slower rate, while *S. oneidensis* carrying OmcA_H10 behaves similarly to that lacking both OmcA and MtrC (Figure 4). This suggests that these mutations may affect the reactivity of the protein with methyl orange.

When evaluating the capacity of the different strains for transferring electrons to an electrode, the behavior was found similar to that obtained with methyl orange. From these assays, it is seen that the SOMR1 ΔOmcA ΔMtrC strain carrying the empty plasmid produced less current when compared with that containing the native OmcA (Appendix A), confirming the importance of OmcA in electron transfer to an electrode. *S. oneidensis* strains carrying OmcA_H1 and OmcA_H3 produce approximately half of the current density obtained for the native protein (Figure 5). This is likely a consequence of the fact that these proteins are less structured, and their amount in *S. oneidensis* cells is significantly lower than for the other mutants or native OmcA (Appendix A).

As observed for methyl orange, the mutation of the axial ligand of heme 10 prevented electron transfer from OmcA to electrodes. Given that this mutation did not affect the growth of the strain during anaerobic conditions with methyl orange (Appendix A), the overall folding of the protein, neither its global redox properties nor its ability to reduce soluble electron shuttles [29], it is possible that the mutation affected the binding process to insoluble electron acceptors, including electrodes, disrupting the electron transfer event. Indeed, heme 10 is at the edge of the protein (see Appendix A) and was proposed to be responsible for the interaction with minerals and metal ions [28].

Interestingly, SOMR1 ΔOmcA ΔMtrC containing OmcA_H6 reduced the electrode at a similar rate as the strain containing the native OmcA (Appendix A). This result is not similar to that observed with methyl orange and suggests that the binding of methyl orange to OmcA occurs near heme 6.

## 4. Discussion

Multiheme cytochromes are key players in EET processes of numerous electroactive organisms. Although amino acid substitutions in these proteins are known to affect protein folding and their mode of action [6,29], these studies have only been performed in vitro. The information on the factors that control electron transfer processes in living organisms is crucial to genetically manipulate them toward improved properties. In this work, we demonstrated that amino acid substitutions can modulate electron transfer, either by changing the redox properties of the protein or by affecting protein folding or the binding process. By replacing the distal axial ligand of each heme of OmcA, we showed that this outer-membrane cytochrome is functionally resilient, and that although some of the mutations affect protein folding and stability, it still sustains the ability of the organisms to perform electron transfer to soluble and insoluble electron acceptors. Among the ten protein mutant variants studied, only the substitution of the distal axial ligand of heme 10, present at the surface of the protein, affected the physiological function of the protein, preventing *S. oneidensis* from transferring electrons to methyl orange and electrodes. Furthermore, the replacement of the histidine of heme 6 with a methionine impacts the electron transfer process to methyl orange but not to electrodes, suggesting that this heme is somehow involved in the specific process of electron transfer to methyl orange. These two observations are in line with the proposal that the staggered cross architecture of OmcA and its homologues is designed to set functional specificity to the various hemes [27].

## Figures and Tables

**Figure 1 microorganisms-11-00079-f001:**
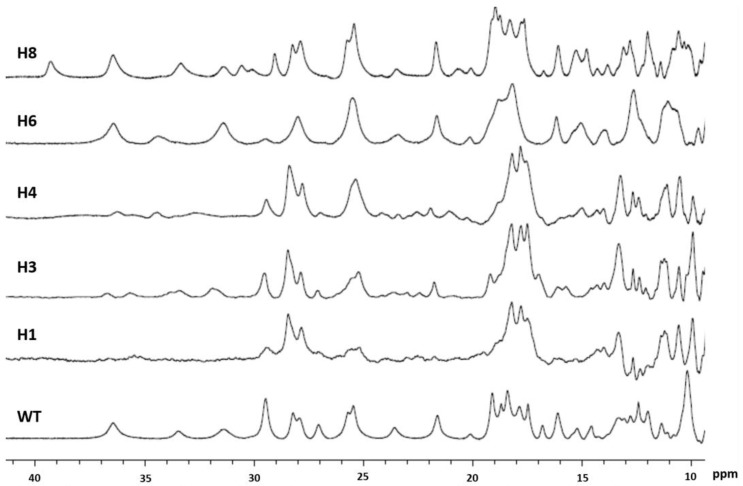
^1^H 1D-NMR spectra of native OmcA and mutants H1, H3, H4, H6, and H8 where the distal axial ligand of the hemes was mutated to a methionine.

**Figure 2 microorganisms-11-00079-f002:**
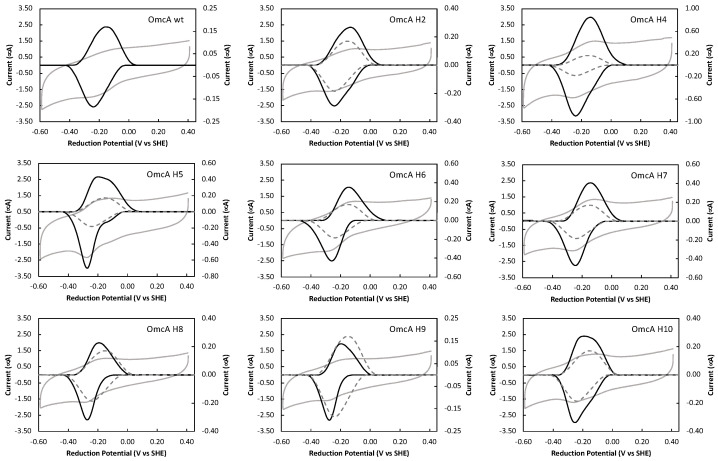
Cyclic voltammetry of native OmcA and eight mutants. In each panel, raw (grey and left axis) and capacitive current subtracted (black and right axis) voltammograms of each OmcA protein (native and mutant) obtained at a scan rate of 100 mV/s are presented. For comparison, the capacitive current-subtracted voltammogram of native OmcA is represented in the panels of the OmcA mutants as dashed line.

**Figure 3 microorganisms-11-00079-f003:**
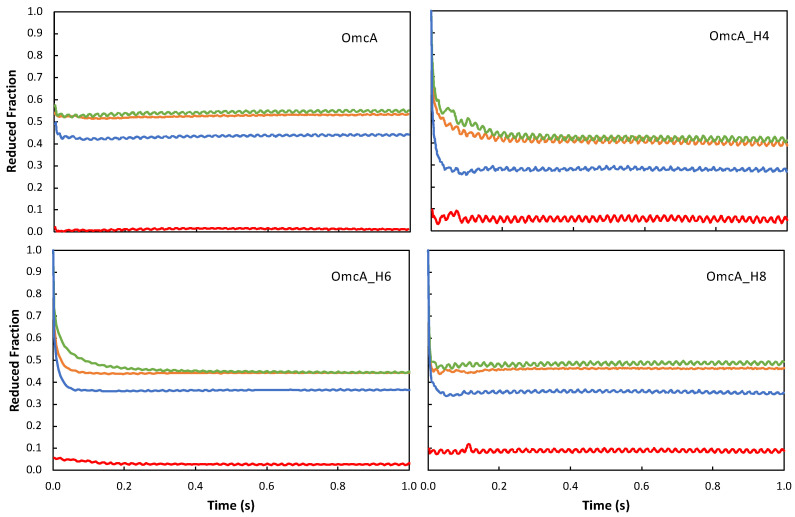
Kinetic traces of oxidation of native OmcA (from [29]) and mutants OmcA_H4, OmcA_H6, and OmcA_H8, by RF (orange), FMN (green), AQDS (blue), and PMS (red). The cytochrome concentration was 0.5 μM, 1.3 μM, and 0.6 μM for mutants OmcA_H4, OmcA_H6, and OmcA_H8, respectively.

**Figure 4 microorganisms-11-00079-f004:**
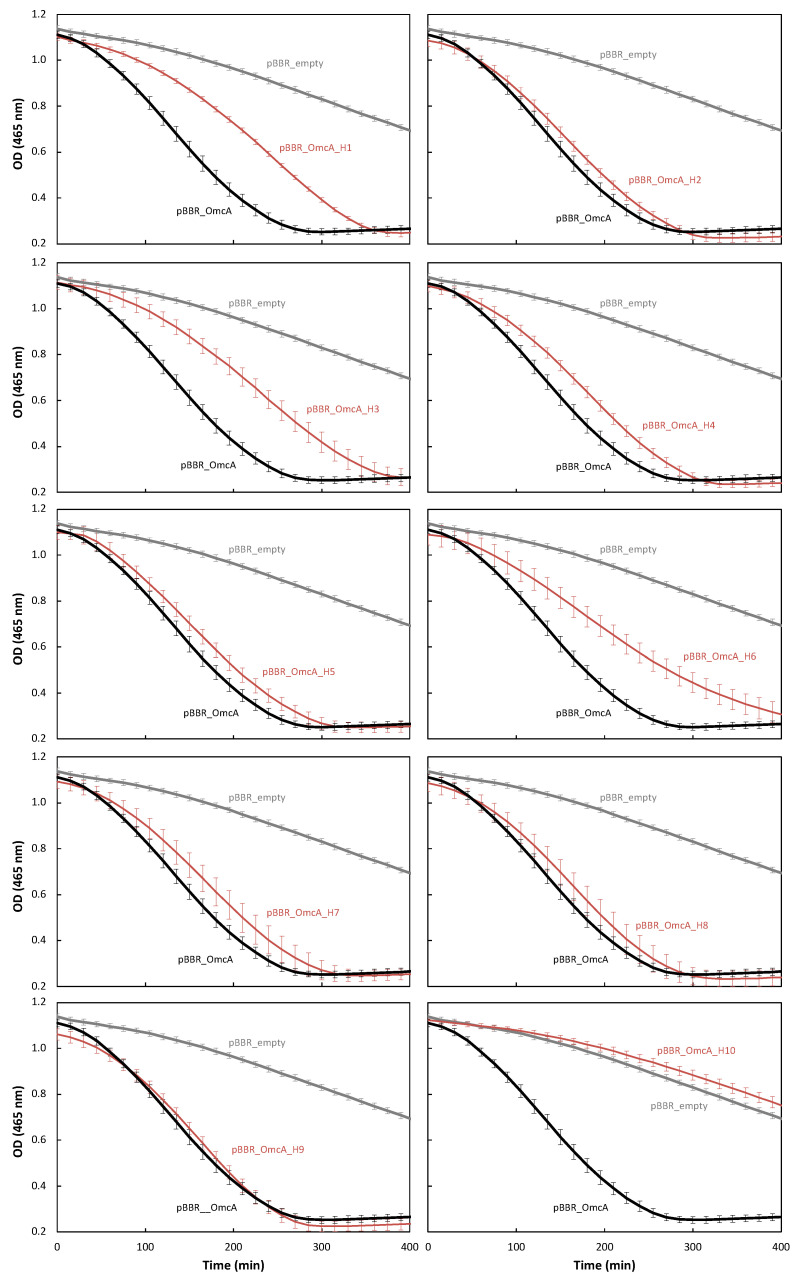
Anaerobic decolorization of methyl orange measured at 465 nm by *S. oneidensis* ΔOmcA ΔMtrC carrying native OmcA: SOMR1 ΔOmcA ΔMtrC/pBBR_OmcA (black line), mutants of OmcA: SOMR1 ΔOmcA ΔMtrC/pBBR_OmcA H1-H10 (red line), and empty plasmid: SOMR1 ΔOmcA ΔMtrC/pBBR_empty (grey line). The error bars represent standard deviations of the measurements.

**Figure 5 microorganisms-11-00079-f005:**
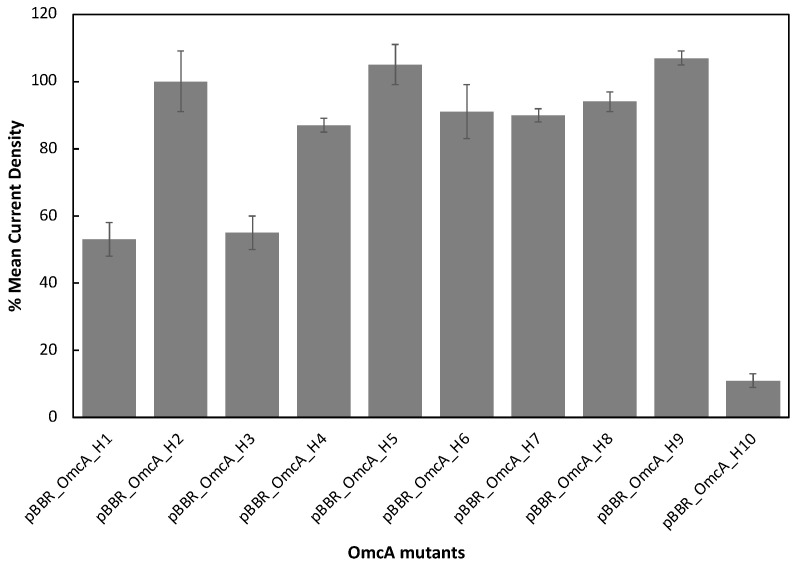
Percentage of the mean current density production of the SOMR1 ΔOmcA ΔMtrC carrying mutants of OmcA: SOMR1 ΔOmcA ΔMtrC /pBBR_OmcA_H1-10, native OmcA: SOMR1 ΔOmcA ΔMtrC/pBBR_OmcA, and the empty plasmid: SOMR1 ΔOmcA ΔMtrC/pBBR_empty. These values were determined using the data obtained by SOMR1 ΔOmcA ΔMtrC/pBBR_OmcA and SOMR1 ΔOmcA ΔMtrC/pBBR_empty as the maximum and minimum current references, respectively.

**Table 1 microorganisms-11-00079-t001:** List of primers used for this work.

Primer	Sequence (5′->3′)
pBBR_OmcA_Forw	GTATCGATAAGCTTGATATCGAAGGAGATATACATACCC
pBBR_OmcA_Rev	CTCTAGAACTAGTGGATCCTTAGTTACCGTGTG
pBBR_OmcA_H9_Rev	CTAGTGGATCCTTAGTTACCCATTGCTTCCATCAATTGCGATG
OmcA_H1_Forw	TACTTATATGATCCATGCTATCCATAAAGGTGGCGAGCGTCA
OmcA_H1_Rev	GAAGGTATGACGCTCGCCACCTTTCATGATAGCATGGATCATATAAGTAAA
OmcA_H3_Forw	TTTACTTATATGATCATGGCTATCCATAAAGGTGGCG
OmcA_H3_Rev	CGCCACCTTTATGGATAGCCATGATCATATAAGTAAA
OmcA_H4_Forw	GGTACGGGAAGTGCAGCTAAACGTCATGGCGATGTAATGAAAG
OmcA_H4_Rev	CTTTCATTACATCGCCATGACGTTTAGCTGCACTTCCCGTACC
OmcA_H6_Forw	CCACGAAAGTGAAGGCATGTATCTGAAATA
OmcA_H6_Rev	TATTTCAGATACATGCCTTCACTTTCGTGG
OmcA_H8_Forw	GCGTGGAAAGCCATGGAAAGTGAAGGCCAT
OmcA_H8_Rev	ATGGCCTTCACTTTCCATGGCTTTCCACGC

**Table 2 microorganisms-11-00079-t002:** List of plasmids used for this work.

Plasmid Name	Gene Variant	Reference
**Plasmid pBBR1MCS-2** [32]	Cell-anchored version	
pBBR_OmcA	Native OmcA	This study
pBBR_empty	(no gene)	This study
pBBR_OmcA_H1	OmcA H277M	This study
pBBR_OmcA_H2	OmcA H240M	This study
pBBR_OmcA_H3	OmcA H274M	This study
pBBR_OmcA_H4	OmcA H384M	This study
pBBR_OmcA_H5	OmcA H359M	This study
pBBR_OmcA_H6	OmcA H618M	This study
pBBR_OmcA_H7	OmcA H576M	This study
pBBR_OmcA_H8	OmcA H613M	This study
pBBR_OmcA_H9	OmcA H733M	This study
pBBR_OmcA_H10	OmcA H696M	This study
**Plasmid pBAD202/D-TOPO**	Soluble version as in [24]	
pBAD_OmcA	Native OmcA	This study
pBAD_OmcA_H1	OmcA H277M	This study
pBAD_OmcA_H2	OmcA H240M	[29]
pBAD_OmcA_H3	OmcA H274M	This study
pBAD_OmcA_H4	OmcA H384M	This study
pBAD_OmcA_H5	OmcA H359M	[29]
pBAD_OmcA_H6	OmcA H618M	This study
pBAD_OmcA_H7	OmcA H576M	[29]
pBAD_OmcA_H8	OmcA H613M	This study
pBAD_OmcA_H9	OmcA H733M	[29]
pBAD_OmcA_H10	OmcA H696M	[29]

**Table 3 microorganisms-11-00079-t003:** Dissociation constants and stoichiometry of binding of FMN by native OmcA and all the OmcA mutants. Values in parenthesis indicate the standard error of the fitted value determined from the diagonal elements of the covariance matrix.

	β_d_ (μM)	n	Reference or Source
**OmcA**	29 (11)	2	[27]
**OmcA_H2**	45 (5)	2	[29]
**OmcA_H4**	116 (6)	2	This work
**OmcA_H5**	169 (22)	2	This work
**OmcA_H6**	135 (22)	2	This work
**OmcA_H7**	137 (26)	2	[29]
**OmcA_H8**	253 (15)	2	This work
**OmcA_H9**	64 (9)	2	This work
**OmcA_H10**	62 (8)	2	This work

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
