# Peer review of "Deciphering Molecular Factors That Affect Electron Transfer at the Cell Surface of Electroactive Bacteria: The Case of OmcA from Shewanella oneidensis MR-1"

_microorganisms, 2022, doi:10.3390/microorganisms11010079_

Round 1

Reviewer 1 Report

See provided peer review docx file

Reviewer 2 Report

The manuscript conducted single amino acid mutations on each heme of the OmcA and studied the properties of the cytochrome mutants as solubility, structural change, redox characteristics, and in-vivo EET capability in engineered S. oneidensis strains. The site-directed mutagenesis method provides a novel insight into the heme functions in terms of EET.  I believe this work is suitable for publication with the minor concerns addressed.

1.     Would the authors have growth curves of the mutant strains harboring different plasmids? Growth conditions would be anaerobic and with electron acceptors such as sodium fumarate or methyl orange, similar to the experiment condition of dye reduction. If the strains grow similarly, one can say for sure the differences in dye reduction were due to cellular EET activities, and not due to cell density differences. For example, the if the strain expressed OmcA_H10 grows slower than the other OmcA mutant strains, then, the lack of dye reduction could also be contributed to the lower number of cells, and thus the lower total number of OmcA_H10. As shown in the absorbance curve in Figure 4, the test time is over 400 min, so the cell growth should be accounted for. 

2.     Page 2, line 63. Please move the full name of AQDS to here as this is the first mention of this chemical. The full name was introduced later on page 3, line 154. 

3.     Page 3, lines 107 and 109. Please double-check the mutant lists from SDM and plasmids of previous work. There seem to be typos as H1, H3, H4, H6, and H8 were developed with SDM, while H2, H5, H7, H9, and H10 were from prior work.

4.     Page 5, line 196. The caption should be in italics and aligned to the left.

Page 7, Figure 2. There are 2 issues with the CV curves, corresponding to Q5 and Q6.

5.     Please change the unit of the left and right axis to more common ones, such as ‘nA’ or ‘mA’. I think the ‘A’ is a typo. 

6.     Please also explain which curves belong to which axes. I would assume the original CV curve (grey) belongs to the left y-axis, while the Faradaic current (black and grey dash) belongs to the right y-axis, but it would be good if the authors could explain the plot.

For the supplementary materials.

7.     Please check the numbering of supplementary figures. There are two Figure S2s.
